# Cohesin and condensin extrude DNA loops in a cell cycle-dependent manner

**Stefan Golfier[1,2,3,4], Thomas Quail[1,2,3,4], Hiroshi Kimura[5], Jan Brugués[1,2,3,4]***

[1]Max Planck Institute of Molecular Cell Biology and Genetics, Dresden, Germany; [2]Max Planck Institute for the Physics of Complex Systems, Dresden, Germany; [3]Centre for Systems Biology Dresden, Dresden, Germany; [4]Cluster of Excellence Physics of Life, TU Dresden, Dresden, Germany; [5]Cell Biology Center, Institute of Innovative Research, Tokyo Institute of Technology, Yokohama, Japan

**Abstract** Loop extrusion by structural maintenance of chromosomes (SMC) complexes has been proposed as a mechanism to organize chromatin in interphase and metaphase. However, the requirements for chromatin organization in these cell cycle phases are different, and it is unknown whether loop extrusion dynamics and the complexes that extrude DNA also differ. Here, we used *Xenopus* egg extracts to reconstitute and image loop extrusion of single DNA molecules during the cell cycle. We show that loops form in both metaphase and interphase, but with distinct dynamic properties. Condensin extrudes DNA loops non-symmetrically in metaphase, whereas cohesin extrudes loops symmetrically in interphase. Our data show that loop extrusion is a general mechanism underlying DNA organization, with dynamic and structural properties that are biochemically regulated during the cell cycle.

## Introduction

Chromatin undergoes a dramatic reorganization during the cell cycle (*Hirano and Mitchison, 1991*; *Rowley and Corces, 2018*; *Nagano et al., 2017*). In interphase, chromatin is organized into compartments and topological-associating domains (TADs) that are cell-type specific (*Bonev and Cavalli, 2016*; *Dekker and Mirny, 2016*; *Rao et al., 2014*). TADs are composed of chromatin loops that have been hypothesized to regulate gene expression by spatially restricting contacts between genes and regulatory elements (*Smith et al., 2016*; *Lupiáñez et al., 2015*; *Ren et al., 2017*; *Schoenfelder and Fraser, 2019*). In metaphase, chromosomes undergo large-scale compaction, leading to the loss of specific boundaries and the shutdown of transcription, which is achieved by arranging chromatin into an array of condensed loops (*Marsden and Laemmli, 1979*; *Earnshaw and Laemmli, 1983*; *Naumova et al., 2013*; *Goloborodko et al., 2016*; *Uhlmann, 2016*; *Kinoshita and Hirano, 2017*). These different degrees of organization require the coordinated activity of protein complexes such as structural maintenance of chromosomes (SMCs) proteins (*Nasmyth, 2001*; *Yatskevich et al., 2019*; *Fudenberg et al., 2016*; *Nuebler et al., 2018*; *Hirano et al., 1997*; *Bouwman and de Laat, 2015*), but how these complexes organize chromatin dynamically during the cell cycle is still unknown. SMCs are thought to organize DNA by actively extruding DNA loops (*Rao et al., 2014*; *Fudenberg et al., 2016*; *Alipour and Marko, 2012*; *Sanborn et al., 2015*). Recent experimental studies have shown that yeast condensin extrudes DNA loops in a one-sided manner *in vitro* (*Ganji et al., 2018*). Although consistent with the loop-extrusion hypothesis, it is inconsistent with the requirement for two-sided loop extrusion predicted by theory (*Banigan and Mirny, 2018*; *Banigan et al., 2019*). One reason for this discrepancy could be that the properties of loop extrusion in cellular contexts differ from those *in vitro* and may be regulated during the cell cycle (*Abramo et al., 2019*; *Losada et al., 1998*). Notably, condensin complexes do not structure the genome during interphase (*Abdennur, 2018*), which raises intriguing questions about the

***For correspondence:**
brugues@mpi-cbg.de

**Competing interests:** The authors declare that no competing interests exist.

molecular players that regulate DNA architecture in interphase. Recent *in vitro* work demonstrated that cohesin can extrude DNA loops symmetrically (*Davidson et al., 2019*; *Kim et al., 2019*), though this activity has not been directly visualized in cellular contexts (*Rao et al., 2017*; *Schwarzer et al., 2017*; *Hansen et al., 2017*). To bridge the gap between *in vitro* biochemical assays and physiological conditions, we used histone H3/H4-depleted *Xenopus laevis* egg extracts to reconstitute loop formation on single DNA molecules. These extracts can be cycled between metaphase and interphase and recapitulate many sub-cellular biological processes, such as the formation of mitotic chromatids and interphase nuclei (*Hirano and Mitchison, 1991*; *Murray, 1991*).

## Results

To visualize DNA loop formation in *Xenopus laevis* egg extracts, we attached lambda-phage DNA to a cover slide using biotin-streptavidin linkers (*Ganji et al., 2016*) in custom-built microfluidic chambers (*Figure 1A*). Addition of either metaphase-arrested or interphase *Xenopus* egg extracts into the chamber triggered the formation of small DNA enrichments, consistent with nucleosomal deposition (*Yan et al., 2007*; *Gruszka et al., 2019*), that rapidly reduced any slack in the DNA molecules (*Figure 1—figure supplement 1A* and *Figure 1—videos 1–2*). To increase the amount of available slack to allow for loop extrusion, we abolished nucleosomal assembly along the strand by depleting ~90–95% of soluble H3-H4 heterodimers in the extract (*Zierhut et al., 2014*; *Figure 1—figure supplement 1B*). This led to the formation of compacted DNA clusters that grew in size over time in both metaphase and interphase (*Figure 1B* and *Videos 1–2*; *Figure 1—videos 3–6*). To investigate whether these clusters exhibited a topology consistent with DNA loops, we hydrodynamically stretched the DNA strand by applying a flow in the perpendicular direction to the strand. This procedure revealed DNA clusters with a characteristic loop topology in both inter- and metaphase extracts (*Figure 1C*, *Figure 1—figure supplement 1C* and *Figure 2—video 1*; *Figure 1—videos 7–9*). In mock-depleted extracts, loops also formed but at a much lower frequency (*Figure 1—figure supplement 1D* and *Figure 1—video 10*) and seemed to compete with nucleosomes for available DNA slack. These results show that DNA loop extrusion can be reconstituted in *Xenopus* egg extracts in metaphase and interphase.

To characterize the dynamic properties of loop formation in *Xenopus* egg extracts, we quantified the DNA distribution inside the loop and to the left and right of the loop as a function of time (*Figure 2*). We computed the loop extrusion rate from the DNA amount that entered the loop over time, and could determine whether this DNA came from one or both of the non-looped regions. Briefly, we summed the fluorescence intensity of the DNA along the perpendicular direction to the DNA strand, and tracked the loop position defined by the local maximum of the DNA intensity. We then fitted a Gaussian function to the loop region and defined the loop boundaries as ±2 standard deviations away from the maximum value of the fit (*Figure 2A*). We obtained the amount of DNA inside the loop as the difference between the integrated intensity in the loop region minus the offset intensity from the Gaussian fit. Finally, the amount of DNA to the left and right of the loop corresponded to the integrated intensity of the DNA strands outside the loop region (see Supplementary Methods). This assay allows us to observe loop extrusion in extract, to quantify the partitioning of DNA between the looped and non-looped regions, and to examine the symmetry of the underlying DNA extrusion process.

When applied to metaphase-arrested, H3-H4-depleted extract (n=7 extract days), this assay showed that DNA loops are initially extruded at 2.36 ± 0.35 kb/s (mean ± SEM) (*Figure 2Bi, C*; *Figure 2—figure supplement 1A*). However, loop growth rapidly slowed down as more DNA was pulled into the looped region (*Figure 2—figure supplement 2A*), suggesting that extrusion rates depend on DNA tension. To examine the relationship between loop extrusion and DNA tension, we used the worm-like chain model (*Marko and Siggia, 1995*) that relates the relative extension of the DNA outside the loop to the corresponding force on the DNA strand (see supplementary methods, *Figure 2—figure supplement 2* and *Figure 2—figure supplement 26*). The relative extension outside the loop is a dynamic quantity defined as: $RE(t)=L/(CL_\lambda-DNA_{loop}(t))$, where L represents the end-to-end binding distance of the DNA molecule on the coverslip, $CL_\lambda$ is the contour length of lambda-phage DNA, and $DNA_{loop}$ is the amount of DNA in the looped region. These results show that the relationship between extrusion rates and DNA tension is generally conserved for all looping events (*Figure 2—figure supplement 2F*). Finally, loop extrusion stopped when the relative extension of

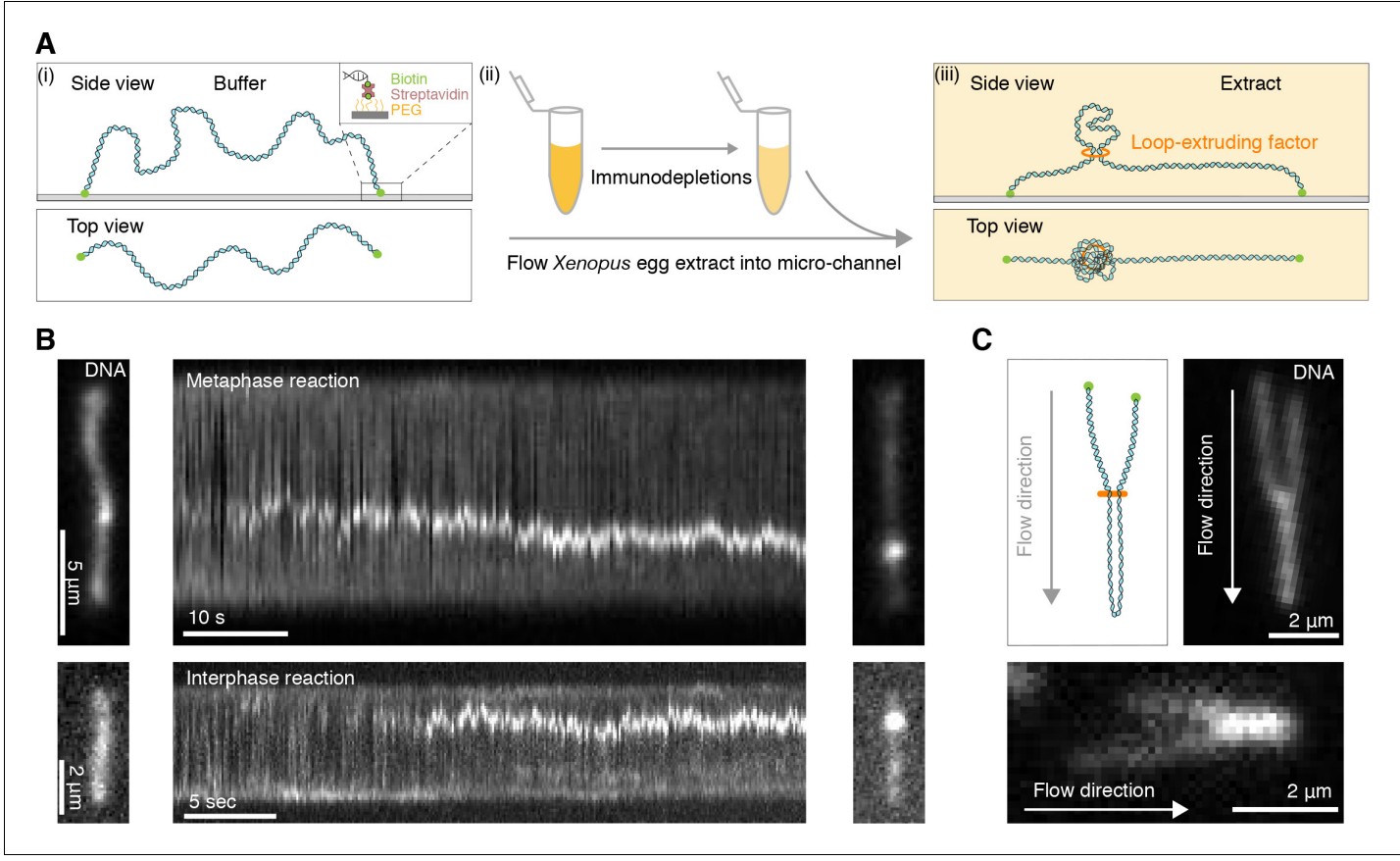

**Figure 1.** Single DNA molecule assay for direct visualization of DNA looping in *Xenopus* egg extracts. (**A**) (**i**) Side and top view schematics of a single strand of λ-phage DNA attached to a functionalized cover slip via biotin-streptavidin linkers. (**ii**) *Xenopus* egg extract is flowed into the microfluidic chamber. (**iii**) Side and top view schematics visualizing how soluble active loop-extruding factors extrude loops in H3-H4-depleted extract. (**B**) Dynamics of the formation of DNA loops induced by H3-H4-depleted extract in metaphase (upper) and interphase (lower). Snapshot of a single molecule of λ-phage DNA visualized using Sytox Orange preceding treatment with H3-H4-depleted extract (left). Kymograph of DNA signal over time displaying a looping event upon addition of H3-H4-depleted extract (middle). Snapshot of steady-state DNA looping event after ~60 s (right). (**C**) Hydrodynamic flows reveal loop topology within DNA cluster. (**i**) Schematic of the loop topology revealed upon flow. (**ii**) Topology of extract-induced DNA loops in metaphase (upper) and interphase (lower) visualized using Sytox Orange revealed upon flow in the direction of the arrow.

The online version of this article includes the following video and figure supplement(s) for figure 1:

**Figure supplement 1.** Characterization of DNA compaction in *Xenopus laevis* egg extracts.

**Figure 1—video 1.** Addition of crude *Xenopus* egg extract to a single strand of λ-phage DNA, visualized using Sytox Orange, leads to the generation of multiple highly-enriched DNA clusters, suggestive of nucleosomal formation along the strand.
https://elifesciences.org/articles/53885#fig1video1

**Figure 1—video 2.** Addition of crude *Xenopus* egg extract to a single strand of λ-phage DNA, visualized using Sytox Orange, leads to the generation of multiple highly-enriched DNA clusters, suggestive of nucleosomal formation along the strand.
https://elifesciences.org/articles/53885#fig1video2

**Figure 1—video 3.** Example of DNA loop formation in H3-H4-depleted egg extract arrested in metaphase visualized using Sytox Orange.
https://elifesciences.org/articles/53885#fig1video3

**Figure 1—video 4.** Example of DNA loop formation in H3-H4-depleted egg extract in metaphase visualized using Sytox Orange.
https://elifesciences.org/articles/53885#fig1video4

**Figure 1—video 5.** Example of DNA loop formation in H3-H4-depleted egg extract in interphase visualized using Sytox Orange.
https://elifesciences.org/articles/53885#fig1video5

**Figure 1—video 6.** Example of DNA loop formation in H3-H4-depleted egg extract in interphase visualized using Sytox Orange.
https://elifesciences.org/articles/53885#fig1video6

**Figure 1—video 7.** Example of hydrodynamically stretched DNA loops in H3-H4-depleted extract arrested in metaphase visualized using Sytox Orange.
https://elifesciences.org/articles/53885#fig1video7

**Figure 1—video 8.** Example of a hydrodynamically stretched DNA loop in H3-H4-depleted interphase extract visualized using Sytox Orange.

*Figure 1 continued on next page*

*Figure 1 continued*

https://elifesciences.org/articles/53885#fig1video8

**Figure 1—video 9.** Example of a hydrodynamically stretched DNA loop in H3-H4-depleted interphase extract visualized using Sytox Orange.

https://elifesciences.org/articles/53885#fig1video9

**Figure 1—video 10.** Example of DNA loop formation in non-depleted crude extract visualized using Sytox Orange.

https://elifesciences.org/articles/53885#fig1video10

DNA outside of the loop reached on average ~65%, corresponding to a stall force of 0.16 pN ± 0.01 pN (mean ± SEM), (*Figure 2D*). In rare cases, we observed individual looping events stalling at DNA extensions of up to ~85%, corresponding to forces up to ~1 pN (*Figure 2—figure supplement 2C–5C*).

To characterize the extrusion symmetry in metaphase, we quantified the total decrease in DNA from the left and right regions of the loop between the onset of loop formation and the final steady-state size of the loop (*Figure 2Bii*). We used these quantities to define a symmetry score as the relative difference between the decrease of these two regions and the total amount of DNA extruded (supplementary methods). The majority of metaphase looping events had a symmetry score close to 1, which corresponds to one-sided (non-symmetric) loop extrusion (*Figure 2Biii*). However, a small population of ~20% of all metaphase looping events were two-sided (as defined by a symmetry score of less than 0.5, *Figure 2—figure supplement 3A*). To complement these symmetry score results, we tracked loop movement along the DNA strand (*Figure 2—figure supplement 4*). Consistent with one-sided loop extrusion, loops that were displaced during loop formation, moved with equal probability towards the boundaries or the center of the strand. This behavior is suggestive of non-symmetric DNA extruding factors landing in a random orientation on the DNA molecule. Taken together, our analysis demonstrates that DNA loop extrusion in metaphase is predominantly one-sided, with extrusion speeds and stall forces similar to those measured *in vitro* (*Ganji et al., 2018*; *Strick et al., 2004*; *Kong et al., 2019*).

Next, we used interphase H3-H4-depleted extract (n = 8 extract days) to investigate whether the dynamics of loop extrusion share similar properties throughout the cell cycle (*Figure 2 Civ-vi*). Loop extrusion in interphase displayed a similar distribution of extrusion rates, with a mean of 1.94 ± 0.26 kb/s, and average stall forces of 0.18 pN ±0.03 pN, with maximal forces of up to 0.82 pN (*Figure 2C and D*, *Figure 2—figure supplement 2F*). However, the distribution of symmetry scores of these looping events peaked towards zero, indicating that these loops are symmetrically extruded. Similar to metaphase looping, we observed that a sub-population of ~20% of all loops had the opposite symmetry (symmetry score larger than 0.5, *Figure 2—figure supplement 3B*). As predicted for symmetric extrusion, loops that started off-center on the strand displayed a strong bias to move towards the DNA boundary of the shorter DNA portion (*Figure 2—figure supplement 4*). Thus, we conclude that the mechanisms of DNA loop extrusion are different in interphase and metaphase.

The different dynamic properties of DNA loop formation that we observe in interphase and metaphase suggest that different molecular activities may be responsible for loop formation during the cell cycle (*Dekker and Mirny, 2016*; *Losada et al., 1998*). The cell cycle-dependent activities of condensin and cohesin could account for the transition between symmetric and non-symmetric loop extrusion (*Banigan et al., 2019*; *Abramo et al.,*

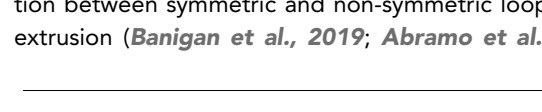

**Video 1.** Example of DNA loop formation in H3-H4-depleted egg extract arrested in metaphase visualized using Sytox Orange. The movie duration is 87 s and the scale bar is 5 μm.

https://elifesciences.org/articles/53885#video1

**Video 2.** Example of DNA loop formation in H3-H4-depleted egg extract in interphase visualized using Sytox Orange. The movie duration is 80 s and the scale bar is 2 μm.

https://elifesciences.org/articles/53885#video2

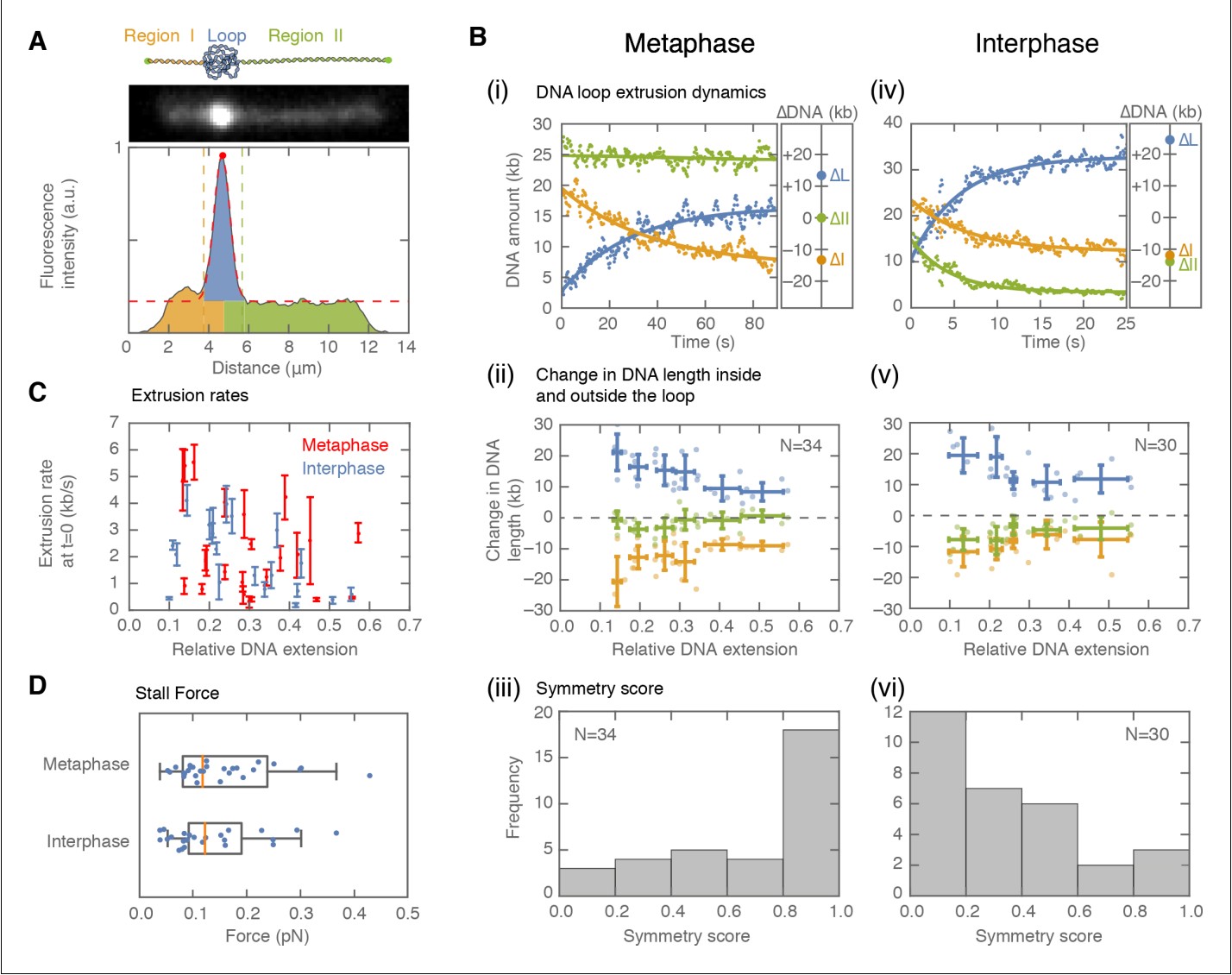

**Figure 2.** Symmetry of DNA loop extrusion is cell cycle-dependent with similar extrusion rates and stalling forces. (A) Method to track DNA-loop dynamics through space and time. *Upper*: Schematic of the top view of a DNA-looping event segmented into three regions: region I (orange), region II (green), and the loop region (blue). *Middle*: Snapshot of DNA-looping event where DNA is labelled using Sytox Orange. *Bottom*: The integrated fluorescence intensity of the DNA generated by summing the intensity values along the perpendicular axis of the strand. The dashed red line represents a Gaussian fit to the data. Signal values above the fit's offset define the looped region given in blue; signal values below this threshold correspond to the non-looped regions I and II, given in orange and in green. To convert the signal into DNA length, the integrated intensity of each region is divided by the total summed intensity of the DNA strand and multiplied by the total length of λ-phage DNA (48.5 kb). (B) Dynamics of DNA looping in H3-H4-depleted extract in metaphase and interphase. (Bi,iv left) DNA amount as a function of time computed for the looped region (blue) and non-looped regions I and II (green and orange). The dots represent experimental data and the solid lines represent exponential fits to the data. (Bi, iv right) The redistribution of DNA during the looping events shown in the left panel quantified as the change in DNA content in each region. (Bii,v) Change in the amount of DNA in the looped region and non-loop regions I and II (as in Bi,iv right) for the entire population of meta- and interphase looping events plotted as a function of the initial relative DNA extension of the corresponding molecules. Error bars correspond to standard deviations of data clustered by proximity. Points represent raw data. (Biii, vi) Analysis of loop extrusion symmetry shows predominantly non-symmetric extrusion (symmetry score ~1) in metaphase (Biii) and symmetric extrusion (symmetry score ~0) in interphase (Bvi). (C) Initial growth rates of DNA loop extrusion in metaphase (red) and interphase (blue) as a function of initial relative DNA extension. These rates were obtained from the slopes of the exponential fits to the loop data at time t = 0 for the subset of loop extrusion events that allowed for a fitting that converged within a tolerance ($10^{-8}$ relative change of the cost function), that corresponded to N = 21 out of 30 in interphase, and N = 24 out of 34 for metaphase. Error bars were obtained from error propagation of the uncertainties of the exponential fit parameters. (D) Box plots of the stall forces for DNA loop extrusion in metaphase and interphase obtained from the final relative extension of the DNA strand at the end of loop formation.

The online version of this article includes the following video and figure supplement(s) for figure 2:

*Figure 2 continued on next page*

*Figure 2 continued*

**Figure supplement 1.** DNA looping examples demonstrating non-symmetric loop extrusion in metaphase and symmetric loop extrusion in interphase.
**Figure supplement 2.** Biophysical characterization of loop extrusion rate and DNA tension.
**Figure supplement 3.** Examples of symmetric DNA loop extrusion in metaphase and non-symmetric loop extrusion in interphase.
**Figure supplement 4.** Characterization of loop movement and symmetry.
**Figure supplement 5.** Dependency of loop extrusion rate on extrusion symmetry and stall forces on initial relative DNA extension.
**Figure supplement 6.** Correction for dye-induced DNA lengthening.
**Figure 2—video 1.** Example of hydrodynamically stretched DNA loops in H3-H4-depleted extract arrested in metaphase visualized using Sytox Orange.
https://elifesciences.org/articles/53885#fig2video1

---

*2019*). To assess the role of cohesin and condensin during loop extrusion in interphase and metaphase, we selectively depleted these protein complexes in *Xenopus* egg extract. We used custom-made antibodies against XSMC1 and XRad21 for cohesin, and XCAP-C and XCAP-E (SMC2 and SMC4) for simultaneous depletion of condensin I and II (*Figure 3—figure supplement 1*). We then tested for loop extrusion activity in each depleted condition. We found that, in metaphase (n = 3 extract days), the occurrence of DNA loop extrusion was significantly reduced (p<0.01) upon depletion of condensin I and II but was unaffected by cohesin depletion (*Figure 3A*). In contrast, there was a significant decrease (p<0.01) in loop extrusion following cohesin depletion in interphase (n = 3 extract days), but was unaffected by condensin depletion (*Figure 3A*). We confirmed these depletions with immunostainings that showed colocalization of cohesin and condensin with the DNA loops observed in interphase and metaphase, respectively (*Figure 3B*). Additionally, we tested for ATPase activity of the loop extrusion factors by enzymatically depleting ATP using apyrase—which for technical reasons was limited to interphase in extract (see supplementary methods). When applied to interphase extract, apyrase-mediated ATP depletion resulted in a near-complete elimination of DNA looping activity , suggesting that cohesin actively extrudes loops in an ATP-dependent manner. Altogether, our results show that cohesin actively extrudes DNA loops symmetrically during interphase, whereas condensin extrudes DNA loops non-symmetrically in metaphase. This demonstrates that the molecular mechanisms of DNA loop extrusion are differentially regulated during the cell cycle.

## Discussion

Our findings provide the first direct evidence that loop extrusion is a general mechanism of DNA organization in a cellular context, and, furthermore, that it is differentially regulated during the cell cycle. This regulation is achieved by the distinct activities of cohesin (*Rao et al., 2017*; *Schwarzer et al., 2017*; *Losada et al., 1998*) and condensin (*Kinoshita and Hirano, 2017*; *Hirano et al., 1997*; *Shintomi et al., 2017*) during interphase and metaphase, and may control different levels of DNA organization during the cell cycle: from chromatin that is mostly decondensed and spatially organized into TAD structures during interphase to highly compacted chromosomes in metaphase (*Abramo et al., 2019*). Symmetric loop extrusion by cohesin in interphase may ensure the formation of specific TAD boundaries by bringing together distal CTCF sites (*Sanborn et al., 2015*; *Tang et al., 2015*). In metaphase, reorganization of loosely packed interphase chromatin into condensed chromosomes leads to the loss of TAD boundaries and the shutdown of transcription (*Nagano et al., 2017*; *Naumova et al., 2013*), which may be achieved by condensin activity (*Goloborodko et al., 2016*). However, many questions remain regarding how the cell cycle regulates condensin and cohesin activities. Previous studies have shown that condensin binds to chromatin in metaphase, but is largely undetected on chromatin in interphase (*Hirano and Mitchison, 1994*); whereas cohesin is bound to chromatin in interphase, but not as strongly in metaphase (*Losada et al., 1998*; *Sumara et al., 2000*). The CDK1-mediated phosphorylation of condensin HEAT subunits in metaphase may be the biochemical signal that triggers the association of condensin to chromatin (*Hirano et al., 1997*). In contrast, most cohesin is released from chromatin by a mechanism that involves the phosphorylation of cohesin's SA subunit (*Losada and Hirano, 2001*; *Hauf et al., 2005*). Thus, the different affinities of condensin and cohesin for chromatin during the

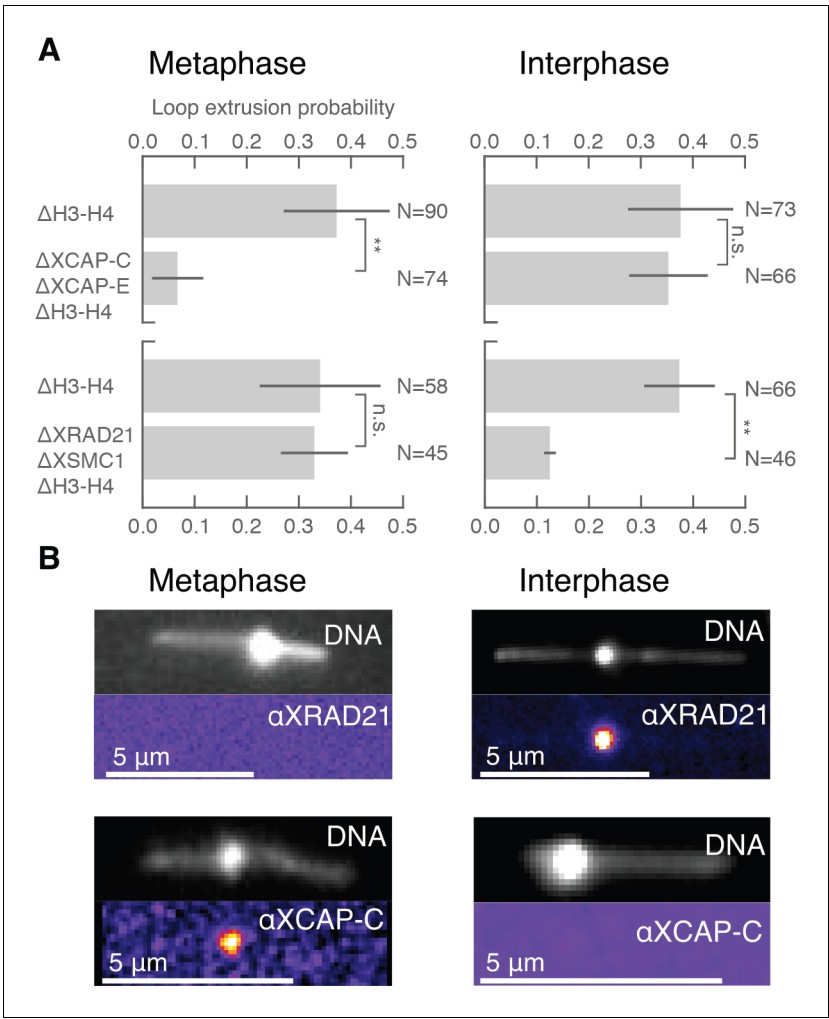

**Figure 3.** Condensin extrudes DNA loops in metaphase and cohesin extrudes loops in interphase. (**A**) DNA loop extrusion probability—the frequency at which looping occurs on a DNA strand with sufficient slack—in metaphase and interphase under different depletion conditions. In metaphase, co-depleting condensin I, condensin II, and H3-H4 (using anti-XCAP-C/E and anti-H4K12Ac) significantly (** represents p<0.01, Binomial test) reduced loop extrusion probability, whereas the same depletion condition in interphase had no effect on loop extrusion probability compared to the control H3-H4-depleted extract. However, co-depleting cohesin and H3-H4 (using anti-XRAD21/XSMC1 and anti-H4K12Ac) had no effect in metaphase, though significantly (p<0.01) decreased loop extrusion probability in interphase compared to H3-H4-depleted extract. (**B**) Snapshots of antibody stainings of representative loops in metaphase and interphase. (Top) In metaphase, Alexa488-labeled anti-XRad21 bound to cohesin does not localize to the DNA loop, whereas in interphase (right panels), the anti-XRad21 co-localizes to the loop. (Bottom) Alexa488-labeled anti-XCAP-C bound to condensin localizes to the DNA loop in metaphase, but does not localize to the loop in interphase.

The online version of this article includes the following figure supplement(s) for figure 3:

**Figure supplement 1.** Co-immunodepletions of *Xenopus* egg extracts using antibodies targeting H3-H4, cohesin, and condensin I and II.

cell cycle could be a natural explanation for the different DNA loop extrusion activities that we see in our experiments.

Our demonstration of predominantly non-symmetric DNA loop extrusion during metaphase is consistent with recent *in vitro* data, but it is at odds with the theoretical requirements to fully compact chromosomes in metaphase (*Banigan and Mirny, 2018*; *Banigan et al., 2019*). However, these studies suggest that a small fraction of two-sided loop extruders—including extrusion events that reel in DNA at different rates from left and right—can facilitate higher levels of chromosome

compaction. Our metaphase data suggest that loops with symmetry scores below 0.8 could be considered 'slow' two-sided extrusion events, as DNA is reeled into the loop from both sides, but at different rates. These events account for about 50% of the total population of metaphase looping events, which, according to theoretical predictions, could be sufficient to achieve 100-fold linear chromosome compaction (*Banigan and Mirny, 2018*). Thus, the mixed populations of loop extrusion symmetries we observe could play a crucial role for proper chromosome organization in metaphase. What is the origin of the small population of symmetric loop extrusion in metaphase? One possibility is that condensin I and condensin II compact DNA using different symmetries (*Kong et al., 2019*). In *Xenopus* egg extract, the relative abundances of condensin I and condensin II is roughly 5:1 (*Ono et al., 2003*), which would be consistent with the fraction of nearly symmetric loop extrusion events that we observe in metaphase (~20%). One limitation of our work, however, is that our antibodies simultaneously depleted those two complexes. As a consequence, we cannot rule out that the small population of symmetric loop extrusion may arise from residual cohesin activity in metaphase. In the future it will be interesting to investigate the origin of the different looping symmetries by using specific antibodies for condensin I and II. In addition, we observe a small sub-population (~20%) of non-symmetric loop extrusion events in interphase, suggesting a differential regulation of extrusion symmetries in both cell cycle phases.

Despite the differences in loop extrusion symmetries between interphase and metaphase, extrusion rates and stall forces seem to be conserved during the cell cycle. The mean extrusion rates we observe, however, are three to four times higher than those observed in previous *in vitro* studies (*Ganji et al., 2018*; *Kim et al., 2019*) for cohesin and condensin respectively. One possibility for this discrepancy could be that, in extract, several extruding factors participate in the extrusion of the same DNA loop in a cooperative manner. However, the average stall forces we estimate are about five to seven times lower than previous estimates *in vitro* (*Ganji et al., 2018*; *Kim et al., 2019*). We speculate that in cytoplasmic context of the H3-H4-depleted egg extract, many other DNA-binding factors—such as linker histone (*Xiao et al., 2012*)—may compete with the loop extrusion machinery for DNA slack. The large spread in the distribution of stall forces, with individual examples reaching values that compare to those reported *in vitro*, may suggest that secondary factors could cause the loop extrusion machinery to stop prematurely, and, consequently, we may underestimate the magnitude of the looping stall forces. We wonder, however, how condensin and cohesin share such similar extrusion rates, even though condensin predominantly extrudes non-symmetrically while cohesin extrudes loops symmetrically. The similar loop growth velocities would suggest that condensin reels in DNA from one side at twice the rate that cohesin reels in DNA from each of its two sides. This assumes, however, that cohesin functions by simultaneously extruding DNA from two sides. Alternatively, cohesin may be a one-sided motor that alternates its extrusion direction (*Banigan et al., 2019*)—though we did not observe this kind of switching within the temporal resolution of our measurements. We speculate that symmetric cohesin loop extrusion could originate from the dimerization of two identical non-symmetric motors, though recent *in vitro* work shows that this idea is controversial (*Davidson et al., 2019*; *Kim et al., 2019*). Interestingly, our results comparing extrusion velocities and corresponding symmetries in interphase show that, on average, symmetric loop extrusion rates are higher (roughly twice) compared to the non-symmetric events (*Figure 2—figure supplement 5A*). This difference in extrusion rates would be consistent with symmetric and non-symmetric loop extrusion mediated by a dimer and a monomer respectively. Our assay will allow for the dissection of the biochemical underpinnings of these processes, and more generally make it possible to reconstitute complex processes such as the formation of boundary elements and the interplay between transcription, replication, and loop extrusion in cellular contexts.

## Materials and methods

### *Xenopus laevis* egg extract preparation, immunodepletions, and ATP depletion

Cytostatic factor (CSF)-arrested *Xenopus laevis* (RRID:XEP_XIa_100) egg extract was prepared as described previously (*Murray, 1991*). In brief, unfertilized oocytes were dejellied and crushed by centrifugation, generating an extract that was arrested in meiosis II. We added protease inhibitors (LPC: leupeptin, pepstatin, chymostatin) and cytochalasin D (CyD) to a final concentration of 10 μg/

ml each to the extract. In order to generate interphase extracts, CaCl$_2$ was added to a final concentration of 0.4 mM. To immunodeplete soluble H3-H4 heterodimers from the extract (*Zierhut et al., 2014*), we coupled 130 µg of a mouse monoclonal anti-H4K12Ac to 12.5 µl rProtein A Sepharose (GE Healthcare) slurry in antibody coupling buffer (10 mM K-HEPES pH = 8, 150 mM NaCl), rotating overnight at 4°C. After several washes with a wash buffer (10 mM HEPES pH = 7.7, 100 mM KCl, 150 mM Sucrose, 1 mM MgCl$_2$), we combined 50 µl fresh CSF extract with the beads and incubated the bead-extract mixture for 1.5 hr on ice, occasionally flicking the tubes in order to prevent the beads settling to the bottom. After recovering the extract from the beads, we immediately proceeded with the experiment. We generated mock-depleted extracts with the same protocol using 130 µg random mouse IgG antibodies (IgG from Mouse (polyclonal)-unconjugated, Jackson Immuno Research) in 50 µl of fresh CSF extract. To co-deplete H3-H4 and both condensin I and condensin II, we coupled 130 µg anti-H4K12Ac and 10 µg rabbit polyclonal antibodies of both anti-XCAP-C and anti-XCAP-E to 15 µl rProtein A Sepharose slurry and performed the same H3-H4 depletion method. To co-deplete H3-H4 and cohesin, we coupled 130 µg anti-H4K12Ac and 10 µg rabbit polyclonal anti-XRad21 and 10 µg anti-XSMC1 to 15 µl rProtein A Sepharose and performed the same H3-H4 depletion method. ATP was depleted by adding 0.03 U/µl apyrase (A6410; Sigma-Aldrich) to the extract reaction in the presence of 5 mM CaCl$_2$, followed by a 15 min incubation at room temperature. The ATP-depleted extract was then introduced into the DNA channels as described below.

## Western blots

We prepared 1:25 dilutions of immunodepleted extract in 1X sample loading buffer (50 mM Tris-HCl, pH = 6.8, 2% SDS, 10% glycerol, 0.006% bromophenol blue, 100 mM DTT), ran a gel electrophoresis on a gradient gel, transferred to a nitrocellulose membrane with a semi-dry transfer approach, and performed primary antibody incubation with polyclonal rabbit antibodies anti-H3 (1:10000, ab1791, RRID:AB_302613), anti-XSMC1 (1:2500, MPI-CBG antibody facility), anti-XCAP-C (1:2000, MPI-CBG antibody facility) and monoclonal mouse antibodies to detect tubulin using anti-DM1a (1:10000, MPI-CBG antibody facility). We detected primary antibodies using LI-COR IRDye secondary antibodies and imaged the western blots using an Odyssey Infrared Imaging System. We analyzed the blots using FIJI.

## Antibody production and labeling

We raised rabbit polyclonal antibodies for immunodepletion against peptides SDIVATPGPRFHTV and DLTKYPDANPNPND corresponding to antibodies that targeted cohesin's XRAD21 and XSMC1 subunits. We also raised rabbit polyclonal antibodies against peptides AAKGLAEMQSVG and SKTKERRNRMEVDK corresponding to antibodies that targeted XCAP-C and XCAP-E for both condensin I and II for immunodepletion (*Hirano et al., 1997*). We added a cysteine residue on the peptide's N-terminus for sulfhydryl coupling, and subsequent keyhole limpet hemocyanin conjugation and affinity purification was performed by MPI-CBG antibody facility. We labeled antibodies with fluorophores for localization using the small-scale on-resin labeling technique from *Groen et al., 2014*. Briefly, we prepared a 200 µl pipette tip to act as our resin bed. We then loaded 40 µl of rProtein A Sepharose (GE Healthcare) resin into the tip, washing three times with 10 mM K-HEPES (pH = 7.7), 150 mM NaCl. We labeled both the antibody targeting the cohesin subunit XRad21 and the antibody targeting condensin I and II's subunit XCAP-C. We flowed 70 µg antibody 5 times consecutively through the packed resin bed in order to bind the antibody to the resin. The resin was then washed three times with 200 mM K-HEPES (pH = 7.7). We then added 0.5 µl 50 mM NHS-Ester-Alexa488 (Alexa Fluor NHS Ester, A20000, Thermo Fischer) to 25 µl 200 mM K-HEPES (pH = 7.7), and immediately added it to the resin, incubating the resin, antibody, and dye for 60 min at room temperature. To remove the unbound dye, the resin bed was washed 5 times with 10 mM K-HEPES (pH = 7.7), 150 mM NaCl. We eluted the labelled antibody with 5 × 15 µl of 200 mM acetic acid. We neutralized each eluate immediately with 5 µl of 1 M Tris-HCl, pH = 9, and cooled to 0°C on ice. The labelled antibody is stable for months kept at 4°C.

## DNA functionalization

To biotinylate DNA purified from lambda-phage (λ-DNA) (*Smith et al., 1996*), we combined 10 µg of λ-DNA (NEB, N3011S) and 5 µl of a 10X polymerase buffer (50 mM Tris-HCl, pH = 7.2, 10 mM

MgSO₄, 100 µM DTT) to a total reaction volume of 50 µl. We then heated the mixture up to 65°C for 7 min to break apart the λ-DNA's sticky ends. After heat treatment, we added 100x molar excess of biotinylated dATP, biotinylated dUTP, and dGTP, and dCTP. We then added one unit (~1 µl) of Klenow enzyme, mixed well, and incubated overnight at room temperature. We purified the biotinylated λ-DNA using ethanol precipitation and stored aliquotes at −20°C.

## PEGylation of cover slips and DNA micro-channel preparation

We functionalized glass cover slips with mPEG and PEG-Biotin. We sonicated coverslips first in acetone for 15 min followed by 5 rinses with MilliQ water, and then another sonication step in 5 M KOH for 40 min. After rinsing the coverslips 3 times with water and then 3 times with methanol, we dried the coverslips with N₂. We silanized the coverslips combining 250 ml methanol, 12.5 ml acetic acid, and 2.5 ml (3-aminopropyl)-trimethoxysilane, incubating the coverslips in this mixture for 10 min at room temperature, sonicating for 1 min, and then incubating the coverslips for an additional 10 min. Next, we rinsed the coverslips once with methanol, once with water, and once again methanol, and dried with N₂. Then we mixed 100 mg mPEG and ~1.5 mg Biotin-PEG with 450 µl PEGylation buffer (0.1M Sodium Bicarbonate, pH = 8.5), and spun the reaction at 10.000 RPM for 1 min to remove air bubbles. We pipetted 25 µl of the PEG mixture onto a dried, silanized coverslip and put another coverslip on top, generating a coverslip sandwich. We incubated these sandwiches over night in distilled water-filled pipette tip-boxes in the dark. After incubation, we carefully disassembled the coverslips, rinsed with MilliQ water, and dried with N₂. To generate a channel for imaging, we first drilled holes through a cleaned cover slide—these holes acted as channel inlets and outlets. We placed custom-designed, laser-cut double-sided tape onto the coverslip, defining the channel geometry. We then placed a functionalized PEG-biotinylated coverslip on top of the double-sided tape, sealing the channel on either end with Valap. We filled the channel with ~10–15 µl of 0.1 mg/ml free streptavidin, incubating the channel with streptavidin for 1 min. To remove the free, unbound streptavidin, we flushed ~100 µl channel washing buffer (40 mM Tris-HCl, pH = 8.0, 20 mM NaCl, 0.4 mM EDTA) through the channel, using the drilled holes as channel inlets and outlets. We added 20 µl of 1:1000 biotinylated λ-DNA (~5 pM), incubating it for ~1 min and then washed the channel with 3 × 100 µl of channel washing buffer.

## Imaging

For live imaging of looping events, we fluorescently stained immobilized DNA strands with 50–500 nM Sytox Orange (S11368, ThermoFisher), a DNA intercalating dye, in imaging buffer (50 mM Tris-HCl pH 7.7, 50 mM KCl, 2.5 mM MgCl₂, 2 mM ATP) similar to *Ganji et al., 2018* or *Xenopus* Buffer (XB: 100 mM KCl, 1 mM MgCl₂, 0.1 mM CaCl₂, 2 mM ATP). We excited Sytox Orange-labelled DNA using a 561 nm laser, and imaged the strands using a Nikon Eclipse microscope stand with a Nikon 100x/NA 1.49 oil SR Apo TIRF and an Andor iXon3 EMCCD camera using a frame-rate of 100–300 ms. A highly inclined and laminated optical sheet (HILO) microscopy mode was established using a Nikon Ti-TIRF-E unit mounted onto the microscope stand to improve signal-to-noise ratio by excluding background fluorescence signal from unbound DNA dye in the buffer. To trigger the formation of DNA loops, we flowed about 2 ul of H3-H4-depleted extract into the channel (total channel volume ~10 ul) and let the extract diffuse further down the channel. We then imaged looping events at the moving front of the diffusing extract. A typical field of view contained 5–20 individual DNA molecules with typically between 2–7 strands having sufficient slack to support loop extrusion. Of these about 30% displayed looping events (*Figure 3*). As we could not control the concentration of loop extrusion factors, the majority of looping events displayed competition between two loops on the same strand. For this study we selected DNA strands that contained only a single looping event per strand.

## Hydrodynamic stretching of loops

To visualize DNA loop topology which cannot be observed in the normal mode of data acquisition, we hydrodynamically stretched DNA strands that exhibited looping events using a flow-controlled syringe pump (Pro Sense B.V., NE-501), see also *Figure 2—video 1*; *Figure 1—videos 7–9*. The flow direction was set to be perpendicular to the strand orientation by a cross-shaped channel design. Depending on the width of the channel, we used flow rates between 100 µl/min and 500 µl/

min to extend DNA loops. Specifically, we introduced H3-H4-depleted extract into the channel as described above and, upon loop formation, stretched DNA strands by flowing imaging buffer from the opposite side.

## Correction of dye-induced DNA lengthening

As mentioned above, we used the DNA intercalating dye Sytox Orange at a range of concentrations to visualize our immobilized lambda DNA molecules. The intercalation of small dye molecules in between adjacent base pairs leads to a dye-concentration-dependent lengthening of the DNA molecules' contour length (*Ganji et al., 2016*). As this effect influences downstream analysis, we sought to correct for the dye-induced lengthening of our DNA molecules by determining the effective contour length of the lambda DNA for each dye concentration used in this study. To this end, we hydrodynamically stretched immobilized DNA molecules in the absence of dye using a buffer flow perpendicular to the strand orientation. DNA molecules were visualized by previous covalently labelling of the DNA backbone with Cy5 fluorescent molecules (label IT nucleic acid labelling kit, Mirus) which does not compromise the DNA contour length. By measuring the extension of these DNA molecules at a certain flow rate, we calculated the corresponding force experienced by the DNA molecules using the worm-like chain model (*Marko and Siggia, 1995*). We then performed the same stretching experiment with DNA molecules exposed to various concentrations of Sytox Orange, keeping the flow rate (and thus the stretching force) constant (1.54 pN) (*Figure 2—figure supplement 6A*). The application of the worm-like chain model to the mean measured DNA extension values for a known force allowed us finally to obtain the contour length of lambda DNA at all examined dye concentrations (*Figure 2—figure supplement 6B*). All these calibrations were done in the same buffer that is present in the channels prior to introduction of the egg extract—which matches the pH and salt concentration of the extract. However, we want to point out, that in the extract, DNA molecules are exposed to a multitude of DNA binding proteins, which may further influence the properties of the DNA. However, for technical reasons, our calibrations of the effect of the dye on DNA length were limited to the buffer condition.

## Loop extrusion analysis

DNA traces were analyzed using custom-written Python scripts motivated by *Ganji et al., 2018*, resulting in data files for further analysis that we added together with the source code in the supplement. We converted movies of fluorescent DNA molecules into one-dimensional intensity profiles by summing the intensity values along the direction perpendicular to the DNA strand in each frame. We removed the background signal using a median filter. From the summed intensity profile for each frame we built kymographs by concatenating all time points (*Figure 2* and *Figure 1—figure supplement 1*). To yield the amount of DNA inside and outside the loop for each time point, we segmented the DNA intensity profiles into a loop region and two regions outside of the loop by first finding the maximum intensity value as the position of the loop and subsequent fitting of a Gaussian around that position. We defined the boundaries of the loop region and the regions outside of the loop by the positions + / - 2 standard deviations from the center of the Gaussian fit. Summing the intensity values of the regions outside of the loop and integrating the intensity under the Gaussian fit yielded the proportions of total signal intensity in each of the three regions for each time point. The difference between the integrated intensity below the loop and the offset from the Gaussian fit (corresponding to the intensity outside of the loop) was equally distributed to the regions outside of the loop as the signal from the incoming and outgoing DNA strands that are not part of the loop itself (*Figure 2A*).

We calculated the relative sizes of the three regions in kilo-base pairs (kb) for each time frame by multiplying the 48.5 kb total length of lambda DNA with the ratio of each summed intensity value and the total summed intensity of the strand for every time point. From these values we calculated the relative change of DNA in each region over time by subtracting the averaged ten last data points from the averaged ten first data points in each region. We used the resulting values $a$ and $b$ for the region left and right of the loop to assign a symmetry score for each looping event by calculating

$$symmetry\ score = \frac{Max(a,b) - Min(a,b)}{a+b}$$

This procedure orders the extrusion from region *a* and *b* such that the symmetry score is always positive and ranges from 0 to 1. Our symmetry score intends to quantify the amount of DNA extruded into the loop from the outer regions. A positive relative change from one side implies that no DNA from that side has been extruded into the loop—and indicates that DNA slipped from the loop to that region— and thus we set that change to 0 (if a > 0: a = 0; if b > 0: b = 0). The slipping of substantial amounts of DNA (>2 kb) was a rare event with three cases in metaphase (p=0.08) and 0 cases in interphase.

This procedure additionally allowed us to track the position of the loop for each time point during every loop extrusion event and study the movement of loops along the DNA strands in inter- and metaphase (*Figure 2—figure supplement 4*). To this end we quantified the change in relative position of the loop by subtracting the average loop positions of the last ten time point from the average loop positions of the first ten time points. (*Figure 2—figure supplement 4C*) This analysis was set up in such a way that, independent of a loop starting left or right of the center of the DNA, the change in loop position was always positive if the loop moved towards or crossed the center of the DNA molecule, and negative if the loop moved towards the closest DNA boundary. The absolute quantity of the change in loop position reflects the relative displacement of the loop along the DNA strand during the process of loop formation and it is referred to as static, if the displacement is below a threshold of 0.08. This analysis allowed us to display the relative displacements of loops as a function of the symmetry score of the corresponding looping in between meta- and interphase (*Figure 2—figure supplement 4D&E*) and compare the probability of the loop to move towards the center of the strand between both cell cycle phases (*Figure 2—figure supplement 4F*).

We extracted the initial loop extrusion rates from the first derivative at time point zero of a single exponential fit to the values of the loop growth over time (*Figure 2B–C*). The size of the loop at each time point further allowed us to continuously calculate the relative extension of the DNA molecule during the loop formation, by dividing the end-to-end distance of the DNA strand by the length of the regions outside of the loop. We estimated the tension on the DNA strand for each time point by applying the Worm Like Chain Model (WLC) of DNA (*Marko and Siggia, 1995*) to these relative extension values (*Figure 2—figure supplement 2C & D*). Since small fluctuations in the estimated relative extension of the DNA, as they occur via thermal agitation of the molecule, can lead to large fluctuations in the corresponding tension, we decided to reduce fluctuations by smoothing the initial loop data. To this end we applied a Savitzky-Golay filter with a 2nd order polynomial and a window size of 63 points to the initial loop data, which significantly reduced fluctuations in the resulting relative DNA extension. The rate of loop extrusion was then extracted from a first order derivative of the smoothed curve and yielded similar initial rates as determined from the exponential fit to the raw loop data. We then applied the WLC model to the smoothed relative extension curve to obtain tension of the DNA molecule for each time point during the loop formation. This procedure allowed us to visualize the decrease in extrusion rate with the increasing tension on the DNA molecule (*Figure 2—figure supplement 2E*). To investigate the dependence of the rate of loop extrusion on the tension on the DNA strand for the entire population of inter- and metaphase looping events, we extracted the extrusion rates and corresponding tension values for each time point during every looping event from the exponential fits to the loop data. This allowed us to display the average decrease in extrusion rate (+ / - standard deviation) for the interphase and metaphase looping populations (*Figure 2—figure supplement 2F*). The stall force of each loop extrusion event (*Figure 2D*) was determined by taking the average steady state loop size of the last ten time points and converting the corresponding relative extension of the DNA molecule into one tension value per looping event using the WLC model. For the analysis of extrusion stall forces we only used DNA strands where the loop extrusion did not end (or was stalled) at the DNA end-binding sites (N = 52).

To quantify the effect of cohesin and condensin depletion, we determined the probability of loop extrusion by counting the number of observable loop extrusion events in all data taken for one condition and dividing it by the total number of DNA strands with sufficient slack (<0.6 relative extension) to support the formation of a loop for that condition.

## Acknowledgements

We acknowledge L Mirny for initial discussions of this work, and AA Hyman, P Tomancak, M Loose, K Ishihara, I Patten, M Sarov, J Guck, F Stewart, F Jülicher, M Srinivasan, and K Nasmyth for

discussions and revision of the manuscript. We thank M Elsner and V Murugesan for assistance in some of the experiments. We thank H Andreas for frog maintenance, the Light Microscopy Facility (LMF), and the Antibody Facility at MPI-CBG.

## Additional information

### Funding

| Funder | Grant reference number | Author |
|---|---|---|
| Human Frontier Science Program | CDA00074/2014 | Jan Brugués |
| European Molecular Biology Organization | ALTF 1456-2015 | Thomas Quail |
| Max-Planck-Gesellschaft | ELBE fellowship | Stefan Golfier |
| Japan Society for the Promotion of Science | JP18H05527 | Hiroshi Kimura |
| Japan Science and Technology Corporation | JPMJCR16G1 | Hiroshi Kimura |

The funders had no role in study design, data collection and interpretation, or the decision to submit the work for publication.

### Author contributions

Stefan Golfier, Conceptualization, Data curation, Software, Formal analysis, Investigation, Methodology, Writing - review and editing; Thomas Quail, Investigation, Methodology, Writing - review and editing; Hiroshi Kimura, Resources, Methodology; Jan Brugués, Conceptualization, Formal analysis, Supervision, Funding acquisition, Writing - original draft, Project administration, Writing - review and editing

### Author ORCIDs

Stefan Golfier  https://orcid.org/0000-0002-4726-667X
Hiroshi Kimura  https://orcid.org/0000-0003-0854-083X
Jan Brugués  https://orcid.org/0000-0002-6731-4130

### Ethics

Animal experimentation: Animal experimentation: All animals were handled according to the directive 2010/63/EU on the protection of animals used for scientific purposes, and the german animal welfare law under the license document number DD24-5131/367/9 from the Landesdirektion Sachsen (Dresden) - Section 24D.

### Decision letter and Author response

Decision letter https://doi.org/10.7554/eLife.53885.sa1
Author response https://doi.org/10.7554/eLife.53885.sa2

## Additional files

### Supplementary files

- Source code 1. DNA-strand analysis.
- Source data 1. Interphase loops raw data.
- Source data 2. Metaphase loops raw data.
- Transparent reporting form

**Data availability**

All data generated or analysed during this study are included in the manuscript.

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
