## [Decision Letter]

**Acceptance summary:**

This work is an important and timely contribution that addresses modalities of loop extrusion by SMC complexes. The present study goes beyond recent published work in this area, by having reconstituted both cohesin and condensin reactions in extracts and having demonstrated the cell cycle dependence of the activity of cohesin and condensin. Presented single-molecule experiments in mitotic and interphase extracts show different modes of loop extrusion, with condensins operating in the mitotic extract as one-sided extruders, and cohesin acting in interphase as a two-sided extruder. Although the process of loop extrusion by yeast condensin has been visualized by single-molecule experiments (Ganji et al., 2018), many questions remain, particularly the mode of loop extrusion by different SMCs complexes of higher eukaryotes. The study of Golfier et al. is the first to examine extrusion by both condensin and cohesin from the same higher organism, in cell extracts, and in a cell-cycle-stage dependent manner. The study also provides important quantitative estimates of the rates and stall forces of extrusion that can be important for future studies aiming to decipher the mechanism of extrusion.

**Decision letter after peer review:**

Thank you for submitting your article "Cohesin and condensin extrude loops in a cell-cycle dependent manner" for consideration by *eLife*. Your article has been reviewed by three peer reviewers, and the evaluation has been overseen by a Reviewing Editor and Kevin Struhl as the Senior Editor. The following individuals involved in review of your submission have agreed to reveal their identity: Leonid A Mirny (Reviewer #1); Andrea Musacchio (Reviewer #2); John F Marko (Reviewer #3).

The reviewers have discussed the reviews with one another and the Reviewing Editor has drafted this decision to help you prepare a revised submission.

Essential revisions:

1) Some caveats need to be included regarding the results. It is emphasized that the experiments put the looping experiments in a cellular context. Yes, extracts are used, but the first result of the paper is that H3-H4 dimers must be removed which makes these experiments rather more like *in vitro* experiments on naked DNA (e.g., by the Dekker and Peters labs) than experiments on chromatin. Further, it is emphasized that the experiments indicate that cohesin carries out symmetric extrusion which condensin is asymmetric, while Figure 2 indicates a mixed distribution, with peaks towards the symmetric and asymmetric cases for interphase/cohesin and metaphase/condensin. According to Figure 2 there is an appreciable subfraction of asymmetric cohesin traces and symmetric condensin traces in the data. The work of Banigan and Mirny puts a 90% compaction limit on pure-one-sided extrusion (with random orientation of extruders) but mixing even a small fraction of two-sided extruders will lead to appreciable further compaction by filling in gaps between the quiet sides of adjacent one-sided extruders.

2) No differentiation is made between condensin I and condensin II; the initial structure of metaphase chromosomes during mitotic prophase is established by condensin II, but *Xenopus* extracts contain condensin I in excess (estimates of Hirano's group are in the 5:1 range for condensin II:condensin I). Have experiments been done along the lines of those in Figure 2 to determine the nature of loop extrusion (symmetric vs. asymmetric) for depletions of condensin I and condensin II-specific components (e.g., XCAP-H and XCAP-H2)? It is therefore not clear whether the two species of condensin have the same loop-extrusion-symmetry properties and it could well be that the more symmetric cases in Figure 2Diii correspond to more symmetric activities of condensin II. For the extracts with excess condensin I it would seem wise to not draw conclusions about condensin II. I do not require new experiments with specific depletions of either condensin I or II (although if they can be added that would be interesting), but do ask for a more careful discussion of the possible roles of the two complexes.

3) There seems to be some inconsistency between the stall forces reported in the text (0.41 pN for condensin, and 0.23pN for cohesin) and those shown in Figure 2D (the mean ~0.1pN for both). I also wonder whether errors in determining the relative extension of DNA could have contributed to the estimate of the stall force. Perhaps authors can test whether estimated stall force for each trajectory depends on the initial and final slack.

4) Looking at Figure 2Biv and v I was a bit surprised that the sum of the length of three regions: DNA to the left from the loop, to the right from the loop, and in the loop don't appear to sum up to a constant. On Figure 2Biv the sum of the three at t=0 is 37Kb, at 30sec its about 43Kb… May be this is just a measurement error… The loop gained 25Kb, but the flanks lost less than 20Kb together. In fitting these curves, I wonder whether the assumption about the sum of the three to be constant was used (and whether using it would make a difference for the rates and the stall forces). Also, in Figure 2Bv, it looks like the loop is getting smaller for trajectories with larger relative extension, (initial, I assume) but the flanks are not getting smaller or larger. This plot is generally hard to understand. Authors may want to have a diagram to help the reader with Figure 2Bii, v and also explain what's shown: one point per trajectory or one point per time from a single (or multiple trajectories).

5) Figure 2C is very important as it provides the estimate of the speed of extrusion. It looks like this plot shows one point (rate at t=0) per trajectory. It may be interesting to see (i) whether rate vs. extension is a universal dependence; and (ii) the extent of variation around this dependence for individual trajectories. Perhaps authors can put not only t=0, but some smoothed velocity vs. extension for different time points on the same plot and/or their exp fits. In other words, if one trajectory starts with less extension and at a higher speed, while another starts at a greater extension and lower speed, will the speed of the first one gets down to the same level when it achieves the same extension as the initial one of the second trajectory. Using these data for all trajectories together (data for different t, not only t=0) may provide a more robust estimate of the rate of extrusion that the two exp fits (one to get rate(t=0) for each trajectory, and the other for rate(t=0) vs. extension).

I wonder whether more accuracy way of measuring the speed of extrusion (or reeling) could be achieved by using very low level of DNA labeling so that individual speckles of DNA can be vitalized and tracked in the flanking regions. This can, in principle, be precise enough to see pausing or switching events. May be this can be tried in future studies.

6) The difference in extrusion dynamics between interphase and mitotic extracts is one of the most exciting aspects of the manuscript. Since cohesin and condensins are labeled it would be great to look closer at the deviations from the textbook picture. So, I wonder whether authors can look at cohesin dynamics in the mitotic extract, and condensin dynamics in the interphase. Are cohesins completely unable to bind or to extrude in the mitotic exact? Can condensins bind/extrude in the interphase, though less efficiently? In other words, are these mere abundances of cohesin in the interphase and condensin in the metaphase, or is there a mechanistic control. Testing for such unconventional roles can be a very exciting twist of the story.

7) The major limit of the study is that there are no mechanistic details to explain the cell-cycle dependency of the reactions. Importantly, DNA loop extrusion by Cohesin had not been demonstrated until very recently, and it is clear that the authors reconstituted this reaction independently from the other two groups. I feel that at least some initial mechanistic detail on how the cell cycle regulates loop extrusion should be discussed.

[Editors' note: further revisions were suggested prior to acceptance, as described below.]

Thank you for re-submitting your article "Cohesin and condensin extrude loops in a cell-cycle dependent manner" for consideration by *eLife*. Your article has been reviewed by three peer reviewers, and the evaluation has been overseen by a Reviewing Editor and Kevin Struhl as the Senior Editor. The following individuals involved in review of your submission have agreed to reveal their identity: Leonid A Mirny (Reviewer #1); Andrea Musacchio (Reviewer #2); John F Marko (Reviewer #3).

The reviewers have discussed the reviews with one another and the Reviewing Editor has drafted this decision to help you prepare a revised submission.

Essential revisions:

1) Reviewer #1 brought up these points:

I found the wording "symmetric" and "non-symmetric loops" a bit confusing, as loops are symmetric, and it is the process that generates them can be either symmetric or non-symmetric. The authors may want to replace "extrudes symmetric loops" with "extrudes loops symmetrically" (and similarly "extrudes loops non-symmetrically") in the Abstract and in the rest of the manuscript.

2) While presenting an equation for the Relative Extension (RE) is really helpful, some inconsistencies between RE presented on different plots remain an issue. Figure 2—figure supplement 5A and B largely disagree with all other plots presenting RE. According to Figure 2—figure supplement 5A, extrusion during interphase results in RE<0.5 for most of the cases, while extrusion during metaphase (Figure 2—figure supplement 5B) appears to be much more processive, resulting in RE>0.5 for most of the cases and at least 8 trajectories showing complete extrusion of all the slack in DNA into a loop (RE=1). Figure 2—figure supplement 5C, on the contrary, has all the points with RE<0.6; including 4 metaphase trajectories with RE<0.2, while only one such point is shown on Figure 2—figure supplement 5B. The number of trajectories (~20) on Figures 2—figure supplement 5A and B also disagree with the numbers of points shown in Figure 2—figure supplement 5C, and all inconsistent with the tallies of trajectories on Figure 2—figure supplement 4D. While Figure 2—figure supplement 5C agrees the main Figure 2, where most of RE <0.6 for both interphase and metaphase, they disagree with the text which states the typical maximum (stalling) RE is ~65% in the metaphase extract (consistent with Figure 2—figure supplement 5B, but not all other figures). Overall, if indeed RE is vastly different in mitotic and interphase trajectories – it's a very interesting result.

3) Since RE was used to infer the stall force, inconsistencies in RE may have propagated to those in the stall force estimates. In general, 0.16-0.18 pN forces reported here (with a smaller stall force for metaphase than for interphase, inconsistent with Figure 2—figure supplement 5C) is a factor of 10 lower than those found by for yeast condensin 1.2+-0.5pN and >0.8pN for human cohesin). It would be great if authors can comment on this. It appears that the quantification of force-extension (Figure 2—figure supplement 6) can make detecting of ~1pN forces extremely unlikely. According to Figure 2—figure supplement 6B, getting a force of about 1pN would require >96% extension. Works of Cees Dekker's lab use magnetic bead quantification (their Nano Letter 2016) where ~1pN are achieved at 80% extension. According to Figure 2—figure supplement 6B, 80% extension gives the force <0.1pN. This factor of 10 difference between force-extension quantification deserves authors attention. Furthermore, Figure 2—figure supplement 2F shows extrusion, albeit at a lower speed, above 0.1pN and up to 1pN force, raising further questions about the estimate of 0.16pN stall force.

4) At the end of the Discussion, the authors bring up a very interesting result (not mentioned in the Results section): "Interestingly, our results comparing interphase extrusion velocities with looping symmetries show that, on average, symmetric loops extrude with a higher rate (roughly twice) compared to the non-symmetric events (Figure 2—figure supplement 5)." Unfortunately, Figure 2—figure supplement 5 doesn't present symmetry vs. rate data. This is very interesting and important result that should be presented in the Results section, and further highlighted in the Discussion. Alongside with the missing symmetry vs. velocity, the authors may want to present symmetry as a function of the initial relative extension, as those slower and one-sided events might reflect partial stalling of the motor activity, and thus may be more abundant at higher extension.

5) The revision presents insightful new analysis of the movement of the loop relative to the border. While interesting, it may be hard for a reader to follow its logic without an illustration (and possibly also a mathematical formula in the Materials and methods). Such a figure can just explain why (i) two-sided extrusion initiated close to the border would make the loop move towards to the nearest border, while those initiated away from the border wouldn't; and (ii) why one-sided extrusion would move a loop to the border or toward the center with 50/50 chances.

6) Sentences comparing one-sided extrusion in metaphase with our recent theory paper are somewhat confusing. The manuscript reads "This (theory) work puts the lower limit on symmetric extrusion at 10%. In contrast to *in vitro* studies of condensin loop extrusion, we found a small sub-population of symmetric loop extrusion in metaphase (~20%) that is above the 10% theoretical limit." Our theory work states that purely one-sided extrusion would leave ~10% of DNA unextruded, setting a limit of ~10-fold compaction due to one-sided extrusion. Hence "10%" in our study does not refer to the fraction of one-sided or two-sided extruders that is required for compaction. However, we did consider mixtures of one-sided and two-sided extruders, demonstrating that a mixture of 40% one-sided and 60% two-sided extruders can achieve 100-fold linear compaction, while achieving 1000-fold compaction would require ~85% two-sided extruders in the mixture. Data presented in the revised manuscript indeed show that in the mitotic extract about 50% of condensins are one-sided (symmetry 0.8-1.0), while the remaining 50% are asymmetric, but two-sided events. As discussed in our PRX paper, even highly asymmetric extrusion can be consistent with "two-sided" extrusion if such asymmetric extruders stay on the chromatin long enough to close the gaps on their "slowly" extruding sides. Thus, the authors can state that this 50% asymmetric extrusion could be sufficient to achieve about 100-fold compaction. Further studies of extruders with different distributions of symmetries and speeds or residence times are underway in our group.

7) Reviewer #3 brings up these points:

The revision has addressed the comments and suggestions of the first round of review. I still think the paper is not as crystal clear as it should be about a key point, namely the use of H3/H4-depleted extracts (without which the effects reported in the paper could not be observed). It should be made very clear that these experiments are not on chromatinized DNA, as most people in the field know is rapidly formed when naked DNA is put into *Xenopus* extracts.

I think that the phrase "histone H3/H4-depleted" should be put in front of "*Xenopus* egg extracts" either in the Abstract or in the last paragraph of the Results.

Numbers of molecules are now included in the Results which is helpful. However, numbers of separate experiments are not listed (N=52 means 52 molecules in one flow cell or 13 molecules observed in 4 separate experiments (flow cells)). Perhaps this information is included in the revision and I apologize if I missed it. However, a clear statement of the number of separate experiments that were used in the study speaks to reproducibility of the results.

---

## [Author Response]

Essential revisions:1) Some caveats need to be included regarding the results. It is emphasized that the experiments put the looping experiments in a cellular context. Yes, extracts are used, but the first result of the paper is that H3-H4 dimers must be removed which makes these experiments rather more like *in vitro* experiments on naked DNA (e.g., by the Dekker and Peters labs) than experiments on chromatin. Further, it is emphasized that the experiments indicate that cohesin carries out symmetric extrusion which condensin is asymmetric, while Figure 2 indicates a mixed distribution, with peaks towards the symmetric and asymmetric cases for interphase/cohesin and metaphase/condensin. According to Figure 2 there is an appreciable subfraction of asymmetric cohesin traces and symmetric condensin traces in the data. The work of Banigan and Mirny puts a 90% compaction limit on pure-one-sided extrusion (with random orientation of extruders) but mixing even a small fraction of two-sided extruders will lead to appreciable further compaction by filling in gaps between the quiet sides of adjacent one-sided extruders.

We thank the reviewers for raising these important points. One of the limitations of our study is that it lacks a comprehensive set of measurements on chromatinized DNA. Indeed we captured a number of looping examples in mock-depleted extracts, though these events were rare. Therefore, we decided to perform most of our measurements using naked DNA and nucleosome-depleted extract. We emphasized this limitation in the manuscript, and hypothesized that the tension generated by nucleosomal incorporation – in mock-depleted extracts – accounted for the low looping efficiency. In future work, our assay can tackle how nucleosomes influence looping activities driven by cohesin and condensin.

As the reviewers point out, Figure 2 displays symmetry scores that reveal differential modes of looping in a cell-cycle dependent manner. In interphase, the symmetry scores peak at “symmetric”, whereas in metaphase the symmetry scores peak at “non-symmetric”. We investigated whether the symmetric looping examples in metaphase (and, in contrast, non-symmetric looping examples in interphase) emerged as a consequence of noisy data, or whether they corresponded to genuine symmetric (or non-symmetric) loops. We now show some of those examples in Figure 2—figure supplement 3, and, indeed, these examples correspond to small sub-populations of symmetric and non-symmetric loops in metaphase and interphase, respectively. We also found that ~20% of all loops in metaphase are symmetric, which is above the 10% limit that Banigan and Mirny require in their paper to fully compact eukaryotic chromosomes. We have added this observation to the paper’s Results section, and comment on how our results relate to the Banigan and Mirny prediction in the Discussion.

2) No differentiation is made between condensin I and condensin II; the initial structure of metaphase chromosomes during mitotic prophase is established by condensin II, but *Xenopus* extracts contain condensin I in excess (estimates of Hirano's group are in the 5:1 range for condensin II:condensin I). Have experiments been done along the lines of those in Figure 2 to determine the nature of loop extrusion (symmetric vs. asymmetric) for depletions of condensin I and condensin II-specific components (e.g., XCAP-H and XCAP-H2)? It is therefore not clear whether the two species of condensin have the same loop-extrusion-symmetry properties and it could well be that the more symmetric cases in Figure 2Diii correspond to more symmetric activities of condensin II. For the extracts with excess condensin I it would seem wise to not draw conclusions about condensin II. I do not require new experiments with specific depletions of either condensin I or II (although if they can be added that would be interesting), but do ask for a more careful discussion of the possible roles of the two complexes.

We immunodepleted condensin using antibodies raised against SMC2 and SMC4, which are subunits shared by both condensin I and II. Thus, as the reviewers point out our study cannot distinguish the possible differential loop extrusion properties of condensin I and II. We have now added this point in the main text.

The Hirano lab estimated the relative abundance of condensin I : condensin II in *Xenopus laevis* egg extract as ~5:1. The Green lab recently showed that condensin I and condensin II compact DNA loops with different symmetries (Kong et al., 2019), which suggests that the symmetric sub-fraction that we observe in our metaphase looping data could originate from a differential looping mechanism of condensin II. To investigate this, we compared the symmetric looping event frequency (symmetry score below 0.5) to the whole population of metaphase looping events, showing that ~ 20% (7 out of 32) of metaphase looping events are symmetric. This number is surprisingly close to the estimated 1:5 relative abundance of condensin II vs. condensin I from the Hirano lab, suggesting a “possible” symmetric mechanism of loop extrusion by condensin II in *Xenopus* egg extract. We plan to perform Condensin I- and Condensin II-specific immunodepletions to understand these differences in symmetry, but these experiments are beyond the scope of this paper.

3) There seems to be some inconsistency between the stall forces reported in the text (0.41 pN for condensin, and 0.23pN for cohesin) and those shown in Figure 2D (the mean ~0.1pN for both). I also wonder whether errors in determining the relative extension of DNA could have contributed to the estimate of the stall force. Perhaps authors can test whether estimated stall force for each trajectory depends on the initial and final slack.

We thank the reviewers for pointing out this typographic error. Instead of 0.41 pN, it should have read 0.14 pN. We have now updated these force values with more data.

We computed the amount of DNA in a loop using relative fluorescence intensities from a region defined as “inside” the loop and two regions “outside” of the loop (Ganji et al., 2018; Ganji et al., 2016). Clearly, this method is vulnerable to DNA signal fluctuations – for example, via thermal fluctuations of the DNA molecule. To address these signal fluctuations, we used stall forces as a measure to describe looping events. Stall forces are more robust than other measurables, such as the initial extrusion rate, because segmentation errors in determining the loop amount decrease as the loop grows. Moreover, since the stall force is calculated when the loop amount reaches a plateau, the fluctuations around this plateau can be removed by fitting, smoothing, or taking a time average. All three methods provide a robust measure of the stall force and give consistent results. We have included these new analyses in Figure 2—figure supplement 2.

As the reviewers suggested, we tested whether the estimated stall forces for each trajectory depend on the initial slack. We found that the stall forces show a slight increase as a function of initial slack, which we now show in Figure 2—figure supplement 5. We reasoned that this slight increase comes from selecting loop extrusion events that have to overcome the initial strand tension. Thus, only loops that have higher stall forces than the initial tension can be measured, leading to a slight increase in the overall stall force as a function of decreasing initial slack. We also show the corresponding initial strand tension as computed from the worm-like-chain model for comparison in Figure 2—figure supplement 5.

4) Looking at Figure 2Biv and v I was a bit surprised that the sum of the length of three regions: DNA to the left from the loop, to the right from the loop, and in the loop don't appear to sum up to a constant. On Figure 2Biv the sum of the three at t=0 is 37Kb, at 30sec its about 43Kb… May be this is just a measurement error… The loop gained 25Kb, but the flanks lost less than 20Kb together. In fitting these curves, I wonder whether the assumption about the sum of the three to be constant was used (and whether using it would make a difference for the rates and the stall forces). Also, in Figure 2Bv, it looks like the loop is getting smaller for trajectories with larger relative extension, (initial, I assume) but the flanks are not getting smaller or larger.

This is an important point, as the total sum of DNA is conserved by definition. The inconsistency in Figure 2B was due to a plotting script error, and has been corrected for the manuscript’s current version. The corresponding stall forces and extrusion rates were not affected as the error occurred in the plotting script.

However, in our previous analysis, we overlooked the fact that the DNA intercalating dye (Sytox orange) lengthens the DNA’s contour length in a concentration-dependent manner. To control for this, we hydrodynamically stretched lambda DNA with a flow rate of known force, and exposed the DNA to buffer containing different Sytox Orange concentrations. By measuring the corresponding length distributions of the DNA molecules for all dye concentrations employed in this study, we acquired a calibration curve that allowed us to determine the contour length of the lambda DNA molecule for each dye concentration. Each looping data set has been re-analyzed with its respective dye-dependent contour length, which slightly affected downstream analysis of the stall forces. We have now added the calibration analysis in the Materials and methods section and showed the calibration curve in Figure 2—figure supplement 6.

This plot is generally hard to understand. Authors may want to have a diagram to help the reader with Figure 2Bii, v and also explain what's shown: one point per trajectory or one point per time from a single (or multiple trajectories).

We agree with the reviewers that the plot is hard to interpret. In Figure 2Bii, v, we sought to demonstrate the change in DNA distribution through the process of loop extrusion for the whole population of loop extrusion events in inter- and metaphase. Every loop extrusion event is represented by three values that display the total change in size of the loop region (blue) and the two regions outside of the loop (green and orange) between the initiation of a loop and the saturation in loop size. We decided to plot those values over the initial extension (the end-to-end distance over the DNA contour length) of the DNA strand to illustrate how those depend on the available slack in the strand. To facilitate the understanding of these plots, we have added small explanatory figures to Figure 2Bi and iv, demonstrating how we convert each individual loop extrusion curve to a triplet of values plotted for the whole population of looping events in Figure 2Bii and v.

5) Figure 2C is very important as it provides the estimate of the speed of extrusion. It looks like this plot shows one point (rate at t=0) per trajectory. It may be interesting to see (i) whether rate vs. extension is a universal dependence; and (ii) the extent of variation around this dependence for individual trajectories. Perhaps authors can put not only t=0, but some smoothed velocity vs. extension for different time points on the same plot and/or their exp fits. In other words, if one trajectory starts with less extension and at a higher speed, while another starts at a greater extension and lower speed, will the speed of the first one gets down to the same level when it achieves the same extension as the initial one of the second trajectory. Using these data for all trajectories together (data for different t, not only t=0) may provide a more robust estimate of the rate of extrusion that the two exp fits (one to get rate(t=0) for each trajectory, and the other for rate(t=0) vs. extension).

We thank the reviewer for this very important suggestion. We have analyzed the tension-dependent extrusion rates in Figure 2—figure supplement 2, including a single trajectory as well as population statistics for inter- and metaphase. Briefly, we analyzed the rate of loop extrusion over time from the smoothed loop data, which we used to further calculate the tension on the DNA molecule for each time point. This allowed us to plot the extrusion rate as a function of DNA tension, but also to demonstrate the effect of small fluctuations in the loop signal on the tension estimate. By fitting the loop data with an exponential function, together with the end-to-end distance of the corresponding DNA strand, we showed the general trend of the extrusion rate as a function of DNA tension for the entire inter- and metaphase population of looping events.

I wonder whether more accuracy way of measuring the speed of extrusion (or reeling) could be achieved by using very low level of DNA labeling so that individual speckles of DNA can be vitalized and tracked in the flanking regions. This can, in principle, be precise enough to see pausing or switching events. May be this can be tried in future studies.

We agree with the reviewer that speckles of DNA would be a complementary method to measure extrusion speed and symmetries. We previously developed a dCas9-EGFP labelling strategy that targeted two specific loci roughly 1/3 and 2/3 along the length of the lamda DNA genome. We consistently observed two GFP spots on the lambda phage genome. However, the low brightness of the probe made it very hard to track these foci on fluctuating DNA strands with enough slack to support loop extrusion. We turned to sparse labelling of the DNA backbone with Cy5 fluorophores, using a commercial kit (Mirus Label IT Nucleic Acid Labelling). This approach gave promising results, and will be optimized for future studies.

Thanks to the suggestion of the reviewer, we developed a complementary approach to identify different loop extrusion symmetries in inter- and metaphase independent of quantifying the DNA fluorescence signal, as shown in Figure 2—figure supplement 4. The method consists of simply tracking the movement of the loop along the DNA strand during its formation. If a non-symmetric loop-extruding motor lands with a random orientation off center of the DNA molecule, it has a 50% chance of reeling-in DNA either from either the short or the long end of the DNA molecule. Consequently, the loop will get pulled either to the center or the boundary of the DNA molecule with equal probabilities. A completely symmetric motor, which reels in DNA from both sides with equal rates, will always pull itself towards the shorter end of the DNA molecule, as the lower amount of total slack in this part will get used up more readily. Consequently, a loop that is formed off-center by a symmetric loop-extruding enzyme, will always move towards the boundary of the DNA molecule. As shown in Figure 2—figure supplement 4, we indeed observe a 50% chance that a loop will move towards the center of the DNA molecule in metaphase, and a much reduced probability to do so in interphase. This suggests a strongly one-sided (non-symmetric) loop extrusion process in metaphase and a two-sided (symmetric) extrusion process in interphase with a certain fraction of non-symmetric cases. We found that there is a strong correlation between asymmetry of loop extrusion and the movement of the DNA loop towards or across the center of the DNA molecule: All of those ‘center-movers’ exhibit strongly non-symmetric DNA redistribution during the formation of the loop. On the other hand, we did not observe a single looping event that exhibited symmetric DNA redistribution and translocated towards the center of the molecule. We have added a short paragraph describing this approach in the Results section.

6) The difference in extrusion dynamics between interphase and mitotic extracts is one of the most exciting aspects of the manuscript. Since cohesin and condensins are labeled it would be great to look closer at the deviations from the textbook picture. So, I wonder whether authors can look at cohesin dynamics in the mitotic extract, and condensin dynamics in the interphase. Are cohesins completely unable to bind or to extrude in the mitotic exact? Can condensins bind/extrude in the interphase, though less efficiently? In other words, are these mere abundances of cohesin in the interphase and condensin in the metaphase, or is there a mechanistic control. Testing for such unconventional roles can be a very exciting twist of the story.

Condensins and cohesins were shown to exhibit tightly regulated alternating chromatin association during the cell cycle (Hirano and Hirano, 1997; Hirano and Mitchison, 1994; Losada and Hirano, 1998). Consistent with this literature, our immunostainings suggest that there is very little DNA association of cohesin in metaphase and no condensin occupancy of DNA in interphase. Unfortunately, we do not have a method of live-imaging cohesin or condensin activity in either of the cell cycles, as all of our antibodies are function-blocking for loop extrusion. We are currently developing strategies to perform these suggested experiments, but these experiments go beyond the scope of the paper. We have now included a discussion of this point in the Discussion section.

7) The major limit of the study is that there are no mechanistic details to explain the cell-cycle dependency of the reactions. Importantly, DNA loop extrusion by Cohesin had not been demonstrated until very recently, and it is clear that the authors reconstituted this reaction independently from the other two groups. I feel that at least some initial mechanistic detail on how the cell cycle regulates loop extrusion should be discussed.

This is an important remark and we have now included a discussion of possible mechanisms of cell cycle regulation of loop extrusion in the Discussion. Generally, the chromosomal association and dissociation of cohesin and condensins shows a strong dependency on the cell cycle with alternating chromatin occupancies by either cohesin in interphase or condensins in metaphase (Losada, Hirano and Hirano, 1997; Abramo et al., 2019). In *Xenopus*, human and mice, about 95% of cohesins dissociate from chromatin upon mitotic entry and only a minor subfraction stays associated with chromatin and is thought to mediate sister chromatid cohesion. The timing of condensin association with chromatin contrasts sharply with cohesin. Condensins are undetectable on interphase chromatin, but bind specifically and abundantly on mitotic chromosomes (Hirano and Mitchison, 1994).

Many questions still remain regarding the mechanisms underlying the cell cycle regulation of cohesin and condensins. For condensins, several mechanisms such as DNA replication, nuclear envelope breakdown or a mutual dependency on cohesin chromatin occupancy can be ruled out (Losada, Hirano and Hirano, 1997). The Hirano group suggests that cdk phosphorylation of one of condensin’s heat subunits could trigger the affinity of condensins for chromatin, as they have observed the hyperphosphorylation of its non-SMC subunits. For cohesin, it has been proposed that the phosphorylation of one of its SA subunits could control its dissociation from chromatin in prophase (Losada and Hirano, 2001). In contrast, CDK1 phosphorylation of soluble cohesin complexes can decrease their ability to bind to chromatin *in vitro*, yet cdk1 activity was not sufficient to dissociate cohesin from chromatin (Losada et al., JCB 2000). On the other hand, the mitosis-specific dissociation of cohesin from chromatin in late prophase in *Xenopus laevis* egg extract was shown to be independent of cyclin B proteolysis and the anaphase promoting complex, suggesting a separase-independent pathway for the bulk of cohesin dissociation from chromatin (Sumara et al., 2000).

[Editors' note: further revisions were suggested prior to acceptance, as described below.]

Essential revisions:1) Reviewer #1 brought up these points:I found the wording "symmetric" and "non-symmetric loops" a bit confusing, as loops are symmetric, and it is the process that generates them can be either symmetric or non-symmetric. The authors may want to replace "extrudes symmetric loops" with "extrudes loops symmetrically" (and similarly "extrudes loops non-symmetrically") in the Abstract and in the rest of the manuscript.

We thank the reviewer for raising this important point. We have changed the wording in the main text as suggested.

2) While presenting an equation for the Relative Extension (RE) is really helpful, some inconsistencies between RE presented on different plots remain an issue. Figure 2—figure supplement 5A and B largely disagree with all other plots presenting RE. According to Figure 2—figure supplement 5A, extrusion during interphase results in RE<0.5 for most of the cases, while extrusion during metaphase (Figure 2—figure supplement 5B) appears to be much more processive, resulting in RE>0.5 for most of the cases and at least 8 trajectories showing complete extrusion of all the slack in DNA into a loop (RE=1). Figure 2—figure supplement 5C, on the contrary, has all the points with RE<0.6; including 4 metaphase trajectories with RE<0.2, while only one such point is shown on Figure 2—figure supplement 5B. The number of trajectories (~20) on Figures 2—figure supplement 5A and B also disagree with the numbers of points shown in Figure 2—figure supplement 5C, and all inconsistent with the tallies of trajectories on Figure 2—figure supplement 4D.

We thank the reviewer for pointing out this error in the caption and x-axis titles of Figure 2—figure supplement 5A and B. The two panels show the initial loop extrusion rate over the symmetry score of the corresponding looping events in inter- and metaphase. Figure 2—figure supplement 5C, in contrast, depicts the stall force of all looping events over the initial relative extension of the DNA molecules. We accidently mislabeled the x-axis in Figure 2—figure supplement 5A and B and propagated this error into the figure legend, albeit correctly referencing it in the main text.

We now corrected the axis labels and figure legends accordingly, indicating the symmetry score of 0 as symmetric and 1 as non-symmetric extrusion in Figure 2—figure supplement 5A and B. As no stable looping events were observed beyond relative extensions of 0.6, we adjusted the range of the x-axis in Figure 2—figure supplement 5C from 0 to 0.8, to make an obvious distinction to the other plots in this figure. We apologize for the confusion.

While Figure 2—figure supplement 5C agrees the main Figure 2, where most of RE <0.6 for both interphase and metaphase, they disagree with the text which states the typical maximum (stalling) RE is ~65% in the metaphase extract (consistent with Figure 2—figure supplement 5B, but not all other figures). Overall, if indeed RE is vastly different in mitotic and interphase trajectories – it's a very interesting result.

The parameter relative extension (RE) always refers to the initial relative extension of the DNA molecule before a loop is formed (i.e. in Figure 2Bii and iv, Figure 2C and Figure 2—figure supplement 5C), as indicated in the figure legends. It must not be confused with the final extension of the DNA molecule at the end of loop extrusion, which is used to calculate the stall force. This can maybe be better understood by looking at Figure 2—figure supplement 2C, showing how the relative extension of one λ DNA molecule increases from an initial value of 0.54 (10um end-to-end distance over 18.5um contour length) prior to loop formation, to a final 0.75 at the end of loop formation.

The relationship between the initial relative extension of the DNA molecule (in other words the distance between the attachment points over its contour length) and the corresponding initial tension on the molecule is shown as a grey dashed line in Figure 2—figure supplement 5C.

In our main figures, we use relative extension as a parameter to show how the size of DNA loops and the extrusion rates decay with increasing end-to-end distance of the molecules, as less slack DNA is available for the loop and higher initial tensions slow the reeling in of DNA (see Figure 2Bii and iv and Figure 2C).

3) Since RE was used to infer the stall force, inconsistencies in RE may have propagated to those in the stall force estimates.

We thank the reviewer for this important point, and assure that the inconsistencies in RE originated exclusively from the error in the x-axis label and caption in Figure 2—figure supplement 5A and B. Stall forces were calculated from the final relative extensions of the DNA molecules at the final steady state of loop extrusion, which usually resulted in final relative extensions around 0.65, corresponding to stall forces below 0.2pN. Only very few examples, as the one shown in Figure 2—figure supplement 2, stalled at higher relative DNA extensions around 0.75.

In general, 0.16-0.18 pN forces reported here (with a smaller stall force for metaphase than for interphase, inconsistent with Figure 2—figure supplement 5C) is a factor of 10 lower than those found by for yeast condensin 1.2+-0.5pN and >0.8pN for human cohesin). It would be great if authors can comment on this.

We now added a discussion of this discrepancy in the main text. In brief, one possibility for the origin of the differences to *in vitro* studies could be, that in extract other DNA-binding proteins, such as linker histone, could bind to the DNA, competing for DNA with the loop extrusion machinery and stalling the loop prematurely. Since we cannot control for these additional factors binding to DNA, we might consequently be underestimating the stall forces of the loop extrusion machinery.

It appears that the quantification of force-extension (Figure 2—figure supplement 6) can make detecting of ~1pN forces extremely unlikely. According to Figure 2—figure supplement 6B, getting a force of about 1pN would require >96% extension. Works of Cees Dekker's lab use magnetic bead quantification (their Nano Letter 2016) where ~1pN are achieved at 80% extension. According to Figure 2—figure supplement 6B, 80% extension gives the force <0.1pN. This factor of 10 difference between force-extension quantification deserves authors attention.

We thank the reviewer for their careful inspection of the manuscript and pointing out this discrepancy. Indeed, we use the same worm-like chain model (Marko and Siggia, 1995) as was employed in the Dekker lab for the aforementioned magnetic bead quantification. The discrepancy was simply caused by an error in the label of the logarithmic y-axis, which was shifted by one order of magnitude. This error most likely happened during the formatting of the fonts in the axis labels. We have corrected this error and apologize for the confusion.

Furthermore, Figure 2—figure supplement 2F shows extrusion, albeit at a lower speed, above 0.1pN and up to 1pN force, raising further questions about the estimate of 0.16pN stall force.

Figure 2—figure supplement 2F was created from the exponential fits to the loop data and relates the extracted extrusion rate to the tension on the DNA molecule, with a bin size of 0.055pN. The reviewer is correct in stating that most of the loops we observe stall at forces below forces of 0.2 pN as shown in Figure 2D. We did however observe three examples of loops that stalled at forces around 0.7pN, as indicated in Figure 2—figure supplement 5C. These constitute the data points in Figure 2—figure supplement 2F that reach up to the 0.8pN bin. The upper bound of 1pN force in this graph was misplaced and we removed it.

4) At the end of the Discussion, the authors bring up a very interesting result (not mentioned in the Results section): "Interestingly, our results comparing interphase extrusion velocities with looping symmetries show that, on average, symmetric loops extrude with a higher rate (roughly twice) compared to the non-symmetric events (Figure 2—figure supplement 5)." Unfortunately, Figure 2—figure supplement 5 doesn't present symmetry vs. rate data.

Please see our response to point 2. As stated there, the x-axis and captions of Figure 2—figure supplement 5A and B were labelled incorrectly and indeed show the extrusion rate over the symmetry score of the corresponding looping event. We have corrected the figure accordingly.

This is very interesting and important result that should be presented in the Results section, and further highlighted in the Discussion.

Indeed, we observe a doubling in extrusion rates when comparing symmetric to non-symmetric looping events in interphase, yet we are limited to very few data points for non-symmetric looping events in that cell cycle phase. Consequently, we bring up this result as an exciting initial observation that we will further investigate in future studies to provide a more comprehensive study of extrusion rates and symmetries.

Alongside with the missing symmetry vs. velocity, the authors may want to present symmetry as a function of the initial relative extension, as those slower and one-sided events might reflect partial stalling of the motor activity, and thus may be more abundant at higher extension.

Upon this interesting suggestion from the reviewer we have analyzed our data in that regard and find no dependency of the loop extrusion symmetry on the initial extension of the DNA molecules.

5) The revision presents insightful new analysis of the movement of the loop relative to the border. While interesting, it may be hard for a reader to follow its logic without an illustration (and possibly also a mathematical formula in the Materials and methods). Such a figure can just explain why (i) two-sided extrusion initiated close to the border would make the loop move towards to the nearest border, while those initiated away from the border wouldn't; and (ii) why one-sided extrusion would move a loop to the border or toward the center with 50/50 chances.

We thank the reviewer for his suggestion and have implemented explanatory figures Figure 2—figure supplement 4A and B which schematically explain the differential movements of loops based on the underlying extrusion symmetry.

6) Sentences comparing one-sided extrusion in metaphase with our recent theory paper are somewhat confusing. The manuscript reads "This (theory) work puts the lower limit on symmetric extrusion at 10%. In contrast to *in vitro* studies of condensin loop extrusion, we found a small sub-population of symmetric loop extrusion in metaphase (~20%) that is above the 10% theoretical limit." Our theory work states that purely one-sided extrusion would leave ~10% of DNA unextruded, setting a limit of ~10-fold compaction due to one-sided extrusion. Hence "10%" in our study does not refer to the fraction of one-sided or two-sided extruders that is required for compaction. However, we did consider mixtures of one-sided and two-sided extruders, demonstrating that a mixture of 40% one-sided and 60% two-sided extruders can achieve 100-fold linear compaction, while achieving 1000-fold compaction would require ~85% two-sided extruders in the mixture. Data presented in the revised manuscript indeed show that in the mitotic extract about 50% of condensins are one-sided (symmetry 0.8-1.0), while the remaining 50% are asymmetric, but two-sided events. As discussed in our PRX paper, even highly asymmetric extrusion can be consistent with "two-sided" extrusion if such asymmetric extruders stay on the chromatin long enough to close the gaps on their "slowly" extruding sides. Thus, the authors can state that this 50% asymmetric extrusion could be sufficient to achieve about 100-fold compaction. Further studies of extruders with different distributions of symmetries and speeds or residence times are underway in our group.

We thank the reviewer for bringing up this a very important point and apologize for the misleading statement in our previous manuscript. We have now updated our discussion of partially symmetric loop extrusion cases in metaphase, according to the suggestions of the reviewer.

7) Reviewer #3 brings up these points:The revision has addressed the comments and suggestions of the first round of review. I still think the paper is not as crystal clear as it should be about a key point, namely the use of H3/H4-depleted extracts (without which the effects reported in the paper could not be observed). It should be made very clear that these experiments are not on chromatinized DNA, as most people in the field know is rapidly formed when naked DNA is put into *Xenopus* extracts.I think that the phrase "histone H3/H4-depleted" should be put in front of "*Xenopus* egg extracts" either in the Abstract or in the last paragraph of the Results.

We thank the reviewer for this suggestion and have changed the manuscript accordingly.

Numbers of molecules are now included in the Results which is helpful. However, numbers of separate experiments are not listed (N=52 means 52 molecules in one flow cell or 13 molecules observed in 4 separate experiments (flow cells)). Perhaps this information is included in the revision and I apologize if I missed it. However, a clear statement of the number of separate experiments that were used in the study speaks to reproducibility of the results.

We have made sure that we state the number of days used to repeat the experiment in the main text and the corresponding figure legends.